# Cadmium-induced impairment in growth, photosynthetic apparatus and redox regulation in green amaranth (*Amaranthus viridis* L.) plant attenuated by salicylic acid and methyl jasmonate

**Md. Tanveer Hussain[1], Md. Sabibul Haque[1]\*, Md. Fazle Rabbi[1], Hafsa Tasnim[1], Md. Asiful Haque[2], Arnab Saha[1], AKM Golam Sarwar[1], Md. Nesar Uddin[1], Md. Alamgir Hossain[1]**

**1** Department of Crop Botany, Bangladesh Agricultural University, Mymensingh-2202, Bangladesh,
**2** Department of Seed Science and Technology, Bangladesh Agricultural University, Mymensingh-2202, Bangladesh

\* mshaqcb@bau.edu.bd

## Abstract

The hazardous Cadmium (Cd) contamination in vegetables from anthropogenic Cd-abundant agroecosystems is a decisive threat to plants and human health. This study examined the prospective roles of salicylic acid (SA) and methyl jasmonate (MeJA) in alleviating Cd-induced toxic effects in green amaranth plants. The seeds of green amaranth (cv. Ghretokanchan) plant were primed with SA (100 and 250 μM) and MeJA (2.5 and 5 μM) and 21-d-old seedlings were set in six conditions under a hydroponic system: (i) Control, (ii) Cd (10 μM $CdCl_2 \cdot H_2O$), (iii) Cd with 100 μM SA, (iv) Cd with 250 μM SA, (v) Cd with 2.5 μM MeJA, and (vi) Cd with 5.0 μM MeJA. The experiment was set in a completely randomized design having three replications. Cd exposure for three weeks markedly impaired plant growth, pigment contents, leaf gas exchange, and photosystem-II efficiency; increased malondialdehyde (MDA) and hydrogen peroxide ($H_2O_2$) levels indicating induced oxidative stress; and enhanced the activities of superoxide dismutase, guaiacol peroxidase and catalase enzymes. However, seed priming and supplementation of SA and MeJA significantly remediated these Cd-mediated adverse effects. Compared to Cd-alone treatment, 100 and 5 μM of SA and MeJA considerably improved total dry weight by 45% and 94%, respectively whereas leaf MDA and $H_2O_2$ levels were substantially reduced by 100 μM SA (37% and 14%) and 5 μM MeJA (35% and 21%). An extensive activity of antioxidants and considerable reduction in Cd uptake and translocation from medium-root-shoot was apparent due to the supplementation of SA and MeJA. The study effectively optimized the levels of SA and MeJA for the improvement of Cd tolerance based on metal uptake, morpho-physiology, and redox regulation, and concludes

**Data availability statement:** All relevant data are within the manuscript itself.

**Funding:** The research was conducted from a research grant supported by the Ministry of Education, the People's Republic of Bangladesh Grant ID: LS20222142; 2022/17/MoE).

**Competing interests:** The authors have no conflict of interest to disclose.

that either 100 µM SA or 5 µM MeJA could be used for the alleviation of Cd-toxicity in green amaranth plants in Cd-contaminated soils through further field trials.

## Introduction

Heavy metal pollution in soils and water bodies due to several anthropogenic events is a global environmental and health concern [1]. Escalated population growth in Bangladesh forces rapid industrialization and increases demand for food production that drives indiscriminate use of heavy metal-containing agrochemicals resulting in heavy metal contamination in farming soils and aquatic ecosystems [2,3]. Heavy metals can be incorporated into the food chain through plant uptake via soil-root-crop or water-root-crop pathways (depending on plant species, the metal and soil conditions) and excess levels of heavy metal in food crops might trigger health hazards for living organisms [4,5]. Among heavy metals, cadmium (Cd) is considered a non-essential and enormous hazardous element for plants, however, it easily enters the plant body from the contaminated soils due to its strong mobility and non-biodegradability [6–8]. The agricultural lands adjacent to industrial areas and roadsides in Bangladesh are reported to be rich in Cd contents [9,10]. Along with these areas, several districts of Bangladesh showed higher Cd contents in soils due to the overuse of chemical pesticides and fertilizers resulting in an increased level of Cd in cereal crops and leafy vegetables including green amaranth (*Amaranthus viridis* L.) [2,11]. The Cd contents in different organs of plants exceed the recommended limit prescribed by the World Health Organization (WHO) and Food and Agriculture Organization (FAO) [12]. Cd exposure to the human body via consumption of Cd-rich plant organs may trigger serious health troubles such as cancer, kidney failure, nephrotoxicity, osteoporosis, respiratory malfunction etc. [13,14]. Leafy vegetables are termed Cd accumulators because of their relatively greater capability of Cd uptake and translocation [15] and these key diets, therefore, play a vital pathway of entering Cd into the human body [16]. The water and mineral nutrients uptake in plants is extensively hindered under Cd-contaminated soils restricting root and shoot elongation and various metabolic activities including respiration rate [17]. Excess Cd in plants slows cell division and growth, induces stomatal closure, and declines chlorophyll synthesis and photosynthesis rate affecting crop yield [5,18–20]. Cd stress hinders the functions of the light-harvesting complex and inhibits photosystem (PS) I and II efficiency limiting photosynthesis [21,22]. A major phenomenon of Cd toxicity is the excess generation of reactive oxygen species (ROS) like singlet oxygen ($^1O_2$), superoxide radicals ($O_2^{\cdot-}$), hydrogen peroxide ($H_2O_2$) and hydroxyl radicals ($OH^\cdot$) triggering rapid oxidation of lipids, proteins and DNA in cells [23,24].

Several approaches can be employed to abate Cd toxicity in crops such as physico-chemical amendments, phytohormone applications, microbe-assisted remediation, organic amendments and genetic strategies [17]. Application of plant growth regulators (PGRs) might be an easy, economical and sustainable strategy to reduce Cd uptake and translocation as well as mitigate the toxic effects of Cd stress in plants [25,26]. Salicylic acid (SA) and methyl jasmonate (MeJA) are two key PGRs and play

a significant role in controlling plant growth and development through signal transduction pathways under Cd stress [27–29]. Exogenous application of SA has been stated to increase growth and photosynthesis, abridge oxidative damage by lowering the production of ROS, electrolytic leakage and malondialdehyde (MDA) contents in crop plants including leafy vegetables [18,22,30]. Salicylic acid supplementation enhanced heavy metal tolerance in plants by regulating the activities of antioxidants, osmolytes levels, metal-chelating compounds and various secondary metabolites [31–33]. SA supplementation has been shown to upregulate the activities of antioxidative enzymes like superoxide dismutase (SOD), catalase (CAT), peroxidase (GPX) and others in many crops balancing redox regulation and boosting resistance to Cd-induced oxidative stress [18,34,35]. A substantial reduction in Cd uptake and translocation was reported due to pretreatment with SA in different cereals [30,36], legumes [37] and leafy vegetables including spinach [22] and menthol mint [31]. Like SA, MeJA on the other hand, mitigates the negative impacts of Cd toxicity in plants by increasing osmolyte concentrations and antioxidant performances that help to cut oxidative damage by lowering the accumulation of MDA, $H_2O_2$ and Cd in plant tissues [38–40]. MeJA modulates plant physiological and biochemical properties under heavy metal stress promoting secondary metabolites synthesis, upregulating the expression of stress-tolerant genes and accelerating the transcriptional activity of genes associated with GSH biosynthesis [41–43]. Several reports highlighted that the exogenous application of MeJA can effectively alleviate Cd uptake, impaired growth and photosynthesis and oxidative damage under Cd stress in *Mentha arvensis* L. [38], rice [44], tomato [40], mustard [45], okra [46], pea [47], wheat [48] and spinach [22] plants.

Leafy vegetables including green amaranth are very popular in Bangladesh due to their short life cycle, ease and low-cost cultivation technique and rich in nutrients. Increasing Cd-contaminated agricultural lands and water bodies is a serious concern for plant growth and human health [9]. Effective strategies should be undertaken for safe vegetable production minimizing the risk to public health. Phytohormone application might be a cost-effective and eco-friendly approach to overcome this issue. Exogenous application of key phytohormones like SA and MeJA for the mitigation of Cd toxicity in leafy vegetables particularly in green amaranth has not yet been documented in Bangladesh. The present study was planned with a hypothesis that supplementation of SA and MeJA with optimum doses will reduce the adverse effects of Cd on green amaranth enabling plant growth, photosynthesis, antioxidative defense and heavy metal tolerance. Therefore, the study designed to examine the potentiality of SA and MeJA in advancing Cd tolerance in green amaranth plants and to explore their proper concentrations in combating Cd uptake and toxicity offering wider applicability in the safe production of green amaranth plants in Cd-affected areas of Bangladesh.

## Materials and methods

### Plant materials and stress conditions

The research was carried out in a hydroponic system at the Crop Botany Department of Bangladesh Agricultural University in Bangladesh between February to April 2024. For this experiment, the green amaranth variety "Ghretokanchan" from a reputed seed company (Lal Teer Seed Ltd., Gazipur, Bangladesh) was used. The availability of green amaranth cultivars is limited in Bangladesh and this variety was chosen among others due to its suitability for round-the-year production, easy availability, fast and vigorous growth, attractive green leaves, softness and deliciousness. Green amaranth seeds were disinfected with Vitavax-200 @1.5 g kg$^{-1}$ seed. It is an effective and worldwide top-grade fungicide (17% Thiram + 17% Carboxin + 66% other ingredients) to prevent seed-borne diseases and is highly recommended by the Bangladesh Agricultural Research Council (BARC) and Bangladesh Agricultural Research Institute (BARI). The seeds were later primed with distilled water for control and Cd-alone treatment while the other four Cd treatments were primed with two concentrations of each of MeJA (2.5 μM and 5 μM) and SA (100 μM and 250 μM) solutions. MeJA (95%) solution was purchased from Sigma-Aldrich, Germany and 11.8 μL was used to prepare a 1 mM stock solution for 50 mL volume. The SA (Sigma Aldrich, Germany) stock solution (1 mM L$^{-1}$) was made by dissolving 138.2 mg in 2 mL methanol and then adding 998 mL distilled water. The seeds were primed by soaking in their respective solutions at a 1:6 seed weight to volume (w/v) proportion and keeping the seeds at 25°C in the dark condition with proper aeration for 12 hours. Primed seeds were

washed for 120 seconds and then dried overnight at room temperature to retain primary moisture levels. Round plastic pots with sterile sand were used for the establishment of seedlings and 14-day-old seedlings were moved to hydroponic tanks (L-11″ × W-7″ × H-7″). Each tank contained 2.5 L of ¼ strength modified Hoagland's nutrient medium [49] maintaining a pH ranging from 5.5 to 6.0 with continuous aeration. After one week, six different growth conditions were implemented under hydroponic system in a controlled growth room, consisting of (i) Control, (ii) Cd (10 µM $CdCl_2 \cdot H_2O$), (iii) $CdSA_{100}$ (Cd with 100 µM SA), (iv) $CdSA_{250}$ (Cd with 250 µM SA), (v) $CdMJ_{2.5}$ (Cd with 2.5 µM MeJA) and (vi) $CdMJ_{5.0}$ (Cd with 5.0 µM MeJA). A completely randomized design with three replications was established. Three tanks (replications) per treatment were maintained having 14 plants in each tank (a total of 42 plants in each treatment). Therefore, A total of 18 hydroponic tanks was set with 252 plants (treatments × replications × plants; 6 × 3 × 14). Cd stress was initiated in hydroponic conditions (21-d old seedlings) by introducing 10 µM $CdCl_2 \cdot H_2O$ in the nutrient solution in all treatments except control and the duration of Cd exposure was three weeks. Consequently, the prescribed doses of SA (100 and 250 µM) and MeJA (2.5 and 5 µM) were mixed along with 10 µM $CdCl_2 \cdot H_2O$ into the nutrient solutions of the respective treatments. The Cd concentration (10 µM $CdCl_2 \cdot H_2O$) was chosen based on a prior trial considering $CdCl_2 \cdot H_2O$ levels between 0–20 µM. The plants could not survive at 20 µM and were not considerably affected while exposed below 5 µM Cd level. As the seedlings were established in two steps (sands and hydroponics) that took a few more days for proper establishment and maintenance, the 21 d-old seedlings were selected for Cd exposure. The total experimental period was six weeks (14 days for seedling establishment, 7 days of regular hydroponic growth and 21 days of Cd treatment). Based on findings from our recent study [22] and supporting literature, we hypothesized that 21 days would be sufficient to elicit measurable responses in leafy vegetables to Cd stress. Additionally, this duration was selected to align with the typical harvest period (40–45 days) for green amaranth. The hydroponic nutrient solutions, including Cd, SA, and MeJA, were replaced every week, with a stepwise rise of the nutrient strength. The plants received artificial illumination from 60 W LED tubes, providing around 250 µmol m$^{-2}$ s$^{-1}$ PPFD (photosynthetic photon flux density) under a 12-h photoperiod. The environmental conditions were adjusted to 25°C (day)/20°C (night), with a relative humidity (RH) of approximately 60–65%.

## Morphological attributes

At 42 DAS, green amaranth plants were harvested, and growth traits in terms of length, fresh and dry mass of root and shoot and leaf area were recorded from four randomly chosen plants of every single treatment. The plant organs such as root, stem and leaves were separated from each plant for morphological data collection. Using a ruler, the longest root length (LRL) and shoot length (SL) were recorded in cm. A digital leaf area meter (LI-3100 C, LI-COR Environmental, Lincoln, USA) was operated to measure the leaf area (LA; cm$^2$ plant$^{-1}$) for each plant. The fresh weight of root (RFW), stem (SFW), leaf (LFW) and their total (TFW) were determined by a digital weighing balance (g plant$^{-1}$). Similarly, the dry mass of root (RDW), stem (SDW), leaf (LDW) and total (TDW) were obtained from the oven-dried (70°C for 72 hours) fresh samples.

## Gas exchange and chlorophyll fluorescence measurements

The rate of photosynthesis (*A*, µmol $CO_2$ m$^{-2}$ s$^{-1}$) was measured in the fully expanded leaves of 42-d old green amaranth plant using a handheld photosynthetic system (LC*i*-SD, ADC Bio Scientific Ltd., Hertfordshire, UK). A portable fluorometer cum porometer (LI-600, LI-COR Environmental, Lincoln, USA) was operated to determine the stomatal conductance ($g_s$, mol m$^{-2}$ s$^{-1}$), rate of transpiration (*E*, mmol m$^{-2}$ s$^{-1}$), quantum yield or operating efficiency of PSII ($\Phi_{PSII}$) and electron transport rate (*ETR*, µmol m$^{-2}$ s$^{-1}$) in the well-developed leaves of three 42-day-old green amaranth plants from each treatment. The maximal quantum efficiency of PSII ($F_v/F_m$) was measured by another fluorometer (Pocket PEA, Hansatech, Norfolk, UK). The selected leaves were dark adapted for 30 min using leaf clips and then $F_v/F_m$ was quantified by applying a saturation pulse of 3500 µmol m$^{-2}$ s$^{-1}$ from the Pocket PEA console. The fluorometer was programmed manually for chlorophyll

fluorescence and water relation traits according to the ambient climatic conditions. Leaf gas exchange traits were measured at a PPFD of nearly 250 μmol m$^{-2}$ s$^{-1}$, ambient air temperature (25°C) and atmospheric $CO_2$ level (400 ppm).

## Leaf Pigment contents

To perform entire biochemical analyses, the leaf samples were acquired on the harvesting day (42 d-old-plant). To quantify leaf Chlorophyll a and b (Chl a and Chl b), total chlorophyll (Total Chl) and total carotenoids (Total Car) contents, nearly 100 mg fresh leaf tissues were subsequently homogenized in 10 mL of acetone (80%), stored for one week in the darkness for ample extraction of the pigments. A UV-vis spectrophotometer (DR6000, Hach, Dusseldorf, Germany) was used to record the optical density at three different wavelengths (470, 646.8 and 663.2 nm) and the pigments were quantified (mg g$^{-1}$ FW) following the principle outlined by Lichtenthaler [50].

## MDA, $H_2O_2$ and proline contents

Malondialdehyde (MDA) content was estimated in the leaves using TBARS (thiobarbituric acid reactive substance) assay [51] with minor alterations [52]. Approximately 100 mg of fresh leaf samples were ground using liquid nitrogen and homogenized with 0.1% TCA (trichloroacetic acid) in a centrifuge tube. After centrifugation (12000 rpm for 15 min), the supernatants were mixed with 0.5% TBA containing 20% TCA in new tubes and kept the tubes in a water bath for 15 min at 90°C. The reactions in the mixtures were stopped immediately by placing the tubes in ice and measured the absorbances at 532 and 600 nm from the samples. The amount of MDA (nmol g$^{-1}$ FW) was calculated by the subtracted absorbance values (532−600) using an extinction coefficient of 155 mM$^{-1}$ cm$^{-1}$ [51]. The $H_2O_2$ content in leaves was measured subsequently following the method suggested by [53]. Briefly, fresh leaf samples of 0.1 g were collected, ground with liquid nitrogen and then homogenate with 0.1% TCA. The reaction mixture of 1.5 mL was prepared including plant extract, 10 mM $KH_2PO_4$ buffer (pH 7.0) and 1 M KI. The composition was kept in the dark for one hour to develop the reaction and then the absorbance was recorded at 390 nm wavelength. For blank, 0.1% TCA without plant extract was used in the assessment. By using a known concentration, a standard curve of $H_2O_2$ was developed to acquire $H_2O_2$ contents (μmol g$^{-1}$ FW) using the absorbance values. With a slight adaptation, the revised acid ninhydrin colorimetric technique recommended by Carillo [54] was used to determine the proline contents of leaves. Nearly 50 mg leaf samples were used to ground using liquid nitrogen, mixed with 70% ethanol and centrifuged. The reaction mixture was formulated containing 1% Ninhydrin (w/v) in 60% acetic acid (v/v) and 20% ethanol (v/v). In a 2 mL Eppendorf tube, 1 mL reaction mixture and 0.5 mL plant extract were mixed and incubated in a water bath for 30 min at 90°C. The samples were cooled at room temperature and taken the absorbance reading at 520 nm from the prepared samples. The proline content (μg g$^{-1}$ FW) was assessed based on the standard curve of L-proline which was developed with known concentrations.

## Antioxidative enzyme activities

From collected leaf tissue samples, roughly 0.1 g was taken to assess the catalytic activity of the SOD (EC1.15.1.1), CAT (EC1.11.1.6) and GPX (EC1.11.1.7). For extraction, samples were crushed using liquid nitrogen in a pre-cooled mortar and pestle and mixed with $KH_2PO_4$ buffer (pH 7.0). Afterwards, the prepared mixture was centrifuged for 20 min at 12,000 rpm maintaining the temperature at 4°C and the extracts were stored at 4°C during the whole analytical procedure. With a trivial amendment, the NBT (nitrobluetetrazolium) colorimetric method proposed by Beauchamp and Fridovich [55] was performed to assess the activity of SOD. In a glass tube, a 3 mL reaction mixture consisting of NBT (75 μM), $KH_2PO_4$ buffer (50 mM, pH 7.8), L-methionine (13 mM), riboflavin (2 μM), EDTA (0.1 mM) and 50 μL of plant extract was prepared. After adding riboflavin at the end, the tubes were placed 30 cm beneath the fluorescent light tubes (2 × 15 W) to trigger the reaction with 15 min of soft shivering. Excluding plant extract, a pair of tubes comprising a reaction mixture was prepared and considered as control (illuminated) and blank (non-illuminated, in dark space). The mixture tubes were placed in a

darkened spot immediately following illumination as soon as a bluish color was formed (highest in control, colorless at a blank, and to a lesser extent bluish in the samples having plant extract than in control). To stop the reaction, the compounds were placed in the dark instantaneously and the optical density was measured at 560 nm. Findings from SOD activity labelled as unit $min^{-1}$ $g^{-1}$ FW and per unit of SOD were exhibited as the magnitude of the enzyme, which hinders NBT photoreduction by 50% per minute. For CAT and GPX extraction, 100 mg of leaf samples were homogenized with $KH_2PO_4$ buffer (pH 7.0) [56]. A volume of 3 mL assay for CAT was prepared consisting of 50 mM $KH_2PO_4$ buffer (pH 7.0), 10 mM $H_2O_2$ and 100 µL sample extract. The changes in absorbance at 240 nm in 30-second intervals (90 sec in total) were observed using a spectrophotometer. The CAT activity (mmol $min^{-1}$ $g^{-1}$ FW)) was then computed using the extinction coefficient of $H_2O_2$ (40 $mM^{-1}$ $cm^{-1}$) as described by Aebi [57]. The GPX assay of 3 mL was made with 50 mM $KH_2PO_4$ buffer (pH 7.0), 2 mM $H_2O_2$, 5 mM Guaiacol and 50 µL sample extract. Leaf extract was added later to initiate the reaction and an increase in absorbance values at 470 nm in 30 sec intervals (90 sec in total) was recorded using the spectrophotometer. The GPX activity (mmol $min^{-1}$ $g^{-1}$ FW) was calculated using the extinction coefficient of Tetraguaiacol (26 $mM^{-1}$ $cm^{-1}$) as prescribed by Fielding and Hall [58].

### Cd contents and translocation factors

The digestion procedure was accomplished by taking 0.5 g ground dried root and leaf samples in a mixture containing nitric acid (20 mL) and perchloric acid (10 mL). Then the digest was placed in a hotplate raising the temperature from 160 to 220°C until the colorless solution was apparent. The absorbance was carried in an AAS (atomic absorption spectrophotometer, AA-7000, Shimadzu, Kyoto, Japan) at 228.8 nm wavelength. The Cd content (µg $g^{-1}$ DW) was quantified using the absorbance values and the standard curve made of pure Cd. The proportion of Cd contents in plant parts to Cd contents in the medium was considered as bioconcentration factor (BCF) while the shoot Cd content to root Cd content was regarded as translocation factor (TF) [59].

### Statistical data analysis

An open-source statistical software 'R' [60] v. 4.0.5 accessed in April 2021 was used to analyze all measured data. Considering treatment as a single factor, analysis of variance (one-way ANOVA) was executed, and the multiple comparisons of treatment means were performed by the Tukey HSD test considering the significance level of $p < 0.05$. From the data, the Stress Tolerance Index (STI) of all measured traits was accounted as (values of Cd stress/values of control) × 100. The standardized STI values of all 30 measured traits were considered to build treatments-traits based hierarchical clustering heatmap following the package *ComplexHeatmap* from the R program. The R packages – *fviz_pca* and *ggplot2* were used to create the principal component analysis (PCA)-biplot.

## Results

### Growth attributes

The growth of green amaranth plants in terms of length, fresh and dry mass of root, stem and leaf were greatly impaired by Cd stress and the traits were notably varied among the treatments (Fig 1, Table 1). The longest root length (LRL) in individual Cd-treated green amaranth plants declined by 36% in comparison to the control. However, LRL reductions were only 12% and 1% in CdSA$_{100}$ and CdMJ$_{5.0}$ treatments, respectively compared to the control. The maximum reduction (60% over control) in shoot length (SL) was found in Cd-alone treatment while the SL was restored in CdSA$_{100}$, CdSA$_{250}$, CdMJ$_{2.5}$ and CdMJ$_{5.0}$ treatments by 88, 16, 54 and 74%, respectively relative to Cd-alone treatment (Table 1, Fig 1). The leaf area (LA) in green amaranth plants significantly declined while exposed to Cd stress without phytohormone supplementation and that reduction was 73% compared to control. In contrast, SA and MeJA supplementation averted Cd-induced leaf area reduction and accounted for 27, 58, 41 and 2% in CdSA$_{100}$, CdSA$_{250}$, CdMJ$_{2.5}$ and CdMJ$_{5.0}$ treatments, respectively

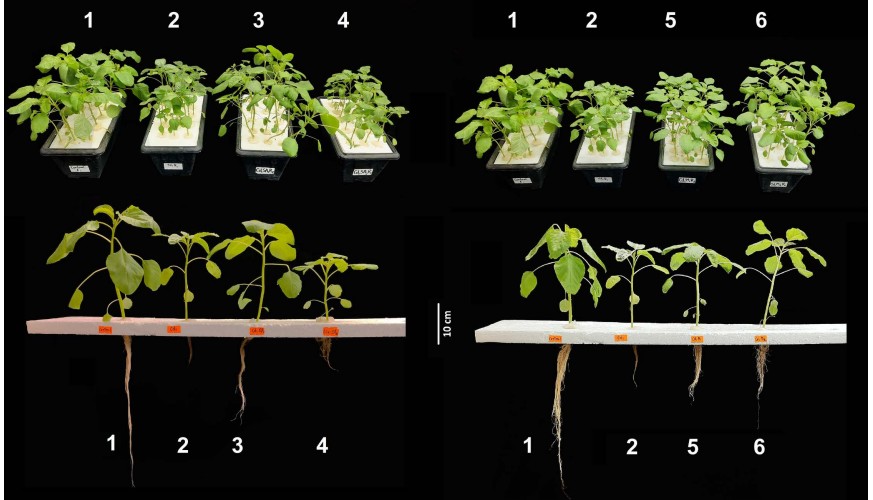

**Fig 1. Growth of 42-day-old green amaranth plants under Cd stress with or without salicylic acid (SA) and methyl jasmonate (MeJA) sup-plementation in hydroponic conditions. (Treatment description- 1: Control; 2: Cd=10µ M CdCl$_2$.** H$_2$O; 3: CdSA$_{100}$=Cd with 100 µM SA; 4: CdSA$_{250}$=Cd with 250 µM SA; 5: CdMJ$_{2.5}$=Cd with 2.5 µM MeJA; 6: CdMJ$_{5.0}$=Cd with 5 µM MeJA).

**Table 1. Growth attributes of 42-day-old green amaranth plants grown in control and Cd with or without SA and MeJA supplementation in hydroponic conditions.**

| Traits | Control | Cd | CdSA$_{100}$ | CdSA$_{250}$ | CdMJ$_{2.5}$ | CdMJ$_{5.0}$ | Sig. level | LSD |
|---|---|---|---|---|---|---|---|---|
| **Longest Root Length** (cm plant$^{-1}$) | 25.8±0.85 a | 10.4±0.99 d | 19.5±0.65 b | 12.0±0.71 d | 16.0±0.41 c | 18.0±0.41 bc | *** | 3.15 |
| **Shoot Length** (cm plant$^{-1}$) | 19.1±0.70 a | 12.2±0.62 b | 16.7±0.63 a | 12.3±0.36 b | 17.2±0.31 a | 19.1±01 a | *** | 2.89 |
| **Leaf Area** (cm$^2$) | 135.6±4.96 a | 36.8±2.74 d | 98.8±8.9 b | 57.5±4.58 cd | 79.6±2.91 bc | 133.2±6.62 a | *** | 24.93 |
| **Root Fresh Weight** (g plant$^{-1}$) | 2.34±0.19 a | 0.46±0.03 d | 1.14±0.12 c | 0.56±0.05 d | 0.92±0.07 cd | 1.64±0.12 b | *** | 0.49 |
| **Stem Fresh Weight** (g plant$^{-1}$) | 2.8±0.09 a | 0.7±0.04 c | 1.7±0.24 b | 0.9±0.05 c | 2.0±0.08 b | 2.9±0.12 a | *** | 0.55 |
| **Leaf Fresh Weight** (g plant$^{-1}$) | 3.6±0.09 b | 1.1±0.13 d | 2.7±0.1 c | 1.6±0.11 d | 3.0±0.19 c | 4.1±0.07 a | *** | 0.53 |
| **Total Fresh Weight** (g plant$^{-1}$) | 8.8±0.25 a | 2.3±0.18 c | 5.6±0.44 b | 3.0±0.19 c | 5.9±0.18 b | 8.7±0.23 a | *** | 1.18 |
| **Root Dry Weight** (mg plant$^{-1}$) | 169±4.8 a | 45±3.2 c | 55±3.9 c | 54±6.8 c | 67±5.1 c | 100±12.4 b | *** | 30.33 |
| **Stem Dry Weight** (mg plant$^{-1}$) | 175±6.7 a | 76±3.4 c | 77±3.6 c | 71±4.7 c | 86±03 c | 129±07 b | *** | 22.44 |
| **Leaf Dry Weight** (mg plant$^{-1}$) | 364±6.1 a | 117±7.3 e | 211±9.1 bc | 142±8.3 de | 173±13.2 cd | 226±14.8 b | *** | 46.29 |
| **Total Dry Weight** (mg plant$^{-1}$) | 708±12.3 a | 237±6.9 d | 343±15.1 c | 266±18.9 cd | 327±21 c | 455±31 b | *** | 85.72 |

Data indicates treatment mean±SEM (n=4). In a row, means with unlike lowercase letters state significant variations corresponding to Tukey's HSD test at $p < 0.05$. ***=0.1% level of significance and LSD stands for Least Significant Difference. Treatment explanation: Cd=10 µM CdCl$_2$.H$_2$O; CdSA$_{100}$=Cd with 100 µM SA; CdSA$_{250}$=Cd with 250 µM SA; CdMJ$_{2.5}$=Cd with 2.5 µM MeJA; CdMJ$_{5.0}$=Cd with 5 µM MeJA.

in relation to control. In Cd-treated plants, the fresh weight of root (RFW), stem (SFW), leaf (LFW) and total (TFW) were considerably reduced by 80, 75, 69 and 74% compared to control while this impairment of fresh weight in all plant organs was restricted in both SA and MeJA-treated plants, particularly in CdSA$_{100}$ and CdMJ$_{5.0}$ (Table 1). Similarly, compared control, the dry weight of root (RDW), stem (SDW), leaf (LDW) and total dry weight (TDW) declined by 74, 57, 68 and 67% due to Cd-alone stress. Compared to individual Cd-treated plants, the TDW was intensified by 45, 12, 48 and 94% in CdSA$_{100}$, CdSA$_{250}$, CdMJ$_{2.5}$ and CdMJ$_{5.0}$, respectively (Table 1). The supplementation of SA and MeJA exhibited dose-dependent attenuation of Cd-induced growth inhibition displaying considerable improvement in 100 µM SA and 5 µM MeJA application.

## Leaf gas exchange and chlorophyll fluorescence parameters

Cd stress significantly impeded the physiological processes in green amaranth plants and the physiological disorders were noticeably restored by the supplementation of SA and MeJA (Table 2). The Cd-alone stress triggered a significant decline in the rate of photosynthesis ($A$) by 40% compared to control leaves, whereas insignificant reductions in $A$ were observed in CdSA$_{100}$ and CdMJ$_{5.0}$ treatments (8% and 5%, respectively) (Table 2). Like $A$, the $g_s$ and $E$ were considerably disturbed by Cd stress accounted to 71% and 63% reductions, respectively compared to control. The $g_s$ and $E$ values in CdMJ$_{5.0}$ were closer to the control while these values were 73% and 75% higher in CdSA$_{100}$ in comparison to Cd-alone treatment, respectively (Table 2). The chlorophyll fluorescence parameters such as $F_v/F_m$, $\Phi_{PSII}$ and $ETR$ were greatly affected due to Cd-alone stress indicating the damage of PSII (Table 2). Cd-alone stress decreased $F_v/F_m$ by 8%, $\Phi_{PSII}$ and $ETR$ by 26% with respect to that of control. Among phytohormone treatments, the CdSA$_{100}$ and CdMJ$_{5.0}$ considerably increased $F_v/F_m$, $\Phi_{PSII}$ and $ETR$ (by 5, 26 and 20% for both the treatments) compared to Cd-alone stress treatment (Table 2).

## Leaf pigment contents

The contents of leaf pigments such as Chl $a$ and $b$, Total Chl and Total Car in green amaranth plants significantly declined in individual Cd-treated plants by 42, 47, 43 and 63%, respectively relative to control (Table 2). Between two levels of SA application, the Chl $a$, Chl $b$, Total Chl and Total Car values increased by 41, 84, 48 and 165% in CdSA$_{100}$ with respect to that of Cd-alone treatment. MeJA-treated leaves showed minimum degradation of leaf pigment contents compared to control (Table 2). The treatment CdMJ$_{5.0}$ enhanced Chl $a$, Chl $b$, Total Chl and Total Car contents by 49, 101, 57 and 142%, respectively in relation to Cd-alone stress treatment (Table 2).

## MDA and $H_2O_2$ contents

Lipid peroxidation, measured as MDA content, was significantly elevated in green amaranth leaves treated with Cd compared to the control, indicating a Cd-mediated oxidative damage (Fig 2A). The extent of increase in MDA in Cd-treated

**Table 2. Leaf gas exchange and chlorophyll fluorescence descriptors and pigment contents of 42-day-old green amaranth plant grown in control and Cd with or without SA and MeJA supplementation in hydroponic conditions.**

| Traits | Control | Cd | CdSA$_{100}$ | CdSA$_{250}$ | CdMJ$_{2.5}$ | CdMJ$_{5.0}$ | Sig. level | LSD |
|---|---|---|---|---|---|---|---|---|
| **Photosynthesis rate** ($A$, µmol m$^{-2}$ s$^{-1}$) | 9.65±0.99 a | 5.84±0.57 bc | 8.89±0.3 ab | 4.9±0.52 c | 7.68±0.44 abc | 9.14±1.43 a | ** | 3.18 |
| **Stomatal Conductance** ($g_s$, mol m$^{-2}$ s$^{-1}$) | 0.134±0.006 a | 0.033±0.004 de | 0.068±0.001 cd | 0.021±0.002 e | 0.083±0.006 bc | 0.108±0.009 ab | *** | 0.03 |
| **Transpiration rate** ($E$, mmol m$^{-2}$ s$^{-1}$) | 1.97±0.072 a | 0.72±0.054 d | 1.26±0.028 c | 0.34±0.025 d | 1.37±0.095 bc | 1.70±0.123 ab | *** | 0.41 |
| **Max photochemical efficiency of PSII** ($F_v/F_m$) | 0.80±0.005 a | 0.73±0.0058 c | 0.77±0.0029 b | 0.73±0.005 c | 0.76±0.0029 b | 0.77±0.0029 b | *** | 0.02 |
| **Quantum yield of PSII** ($\Phi_{PSII}$) | 0.70±0.014 a | 0.52±0.0037 c | 0.66±0.004 ab | 0.54±0.0108 c | 0.59±0.021 bc | 0.66±0.021 ab | *** | 0.07 |
| **Electron Transport Rate** ($ETR$, µmol m$^{-2}$ s$^{-1}$) | 73.6±1.47 a | 54.8±0.39 c | 68.9±0.40 ab | 56.8±1.14 c | 62.4±2.17 bc | 69.0±2.16 ab | *** | 8.12 |
| **Chlorophyll $a$** (mg g$^{-1}$ FW) | 2.43±0.17 a | 1.41±0.12 b | 2.28±0.022 a | 1.98±0.07 ab | 1.87±0.19 ab | 2.09±0.06 a | ** | 0.67 |
| **Chlorophyll $b$** (mg g$^{-1}$ FW) | 0.50±0.049 a | 0.27±0.022 b | 0.46±0.014 ab | 0.49±0.02 a | 0.42±0.052 ab | 0.53±0.048 a | * | 0.20 |
| **Total Chlorophyll** (mg g$^{-1}$ FW) | 2.93±0.2172 a | 1.67±0.12 b | 2.74±0.032 a | 2.47±0.09 ab | 2.30±0.24 ab | 2.63±0.10 a | ** | 0.84 |
| **Total Carotenoids** (mg g$^{-1}$ FW) | 3.46±0.196 a | 1.30±0.060 b | 3.79±0.302 | 3.43±0.182 a | 2.90±0.151 a | 3.14±0.393 a | *** | 1.31 |

Data indicates treatment mean±SEM (n=3). In a row, means with unlike lowercase letters state significant variations corresponding to Tukey's HSD test at $p<0.05$. ***, ** and * reflect 0.1%, 1% and 5% levels of significance, respectively. LSD stands for Least Significant Difference. Treatment explanation: Cd = 10 µM CdCl$_2$.H$_2$O; CdSA$_{100}$ = Cd with 100 µM SA; CdSA$_{250}$ = Cd with 250 µM SA; CdMJ$_{2.5}$ = Cd with 2.5 µM MeJA; CdMJ$_{5.0}$ = Cd with 5 µM MeJA.

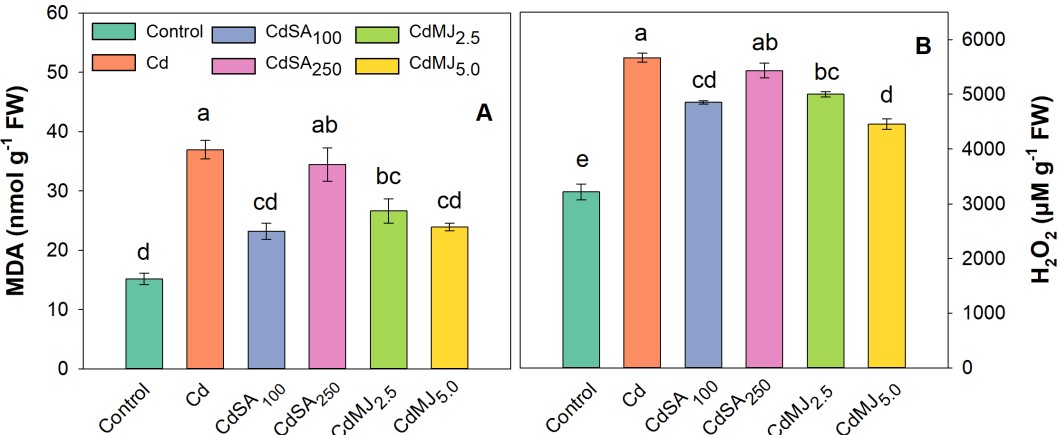

**Fig 2. Leaf Malondialdehyde (MDA; A) and Hydrogen Peroxide ($H_2O_2$; B) contents in 42-day-old green amaranth plants grown in control and Cd with or without SA and MeJA supplementation in hydroponic conditions.** The upright bars relate to SEM (n = 3). Treatment means with unlike lowercase letters within the treatments denote significant variations corresponding to Tukey's HSD test at $p < 0.05$. Treatment explanation: Cd = 10 μM $CdCl_2.H_2O$; $CdSA_{100}$ = Cd with 100 μM SA; $CdSA_{250}$ = Cd with 250 μM SA; $CdMJ_{2.5}$ = Cd with 2.5 μM MeJA; $CdMJ_{5.0}$ = Cd with 5 μM MeJA.

leaves was 144% than control, while these increase over control in $CdSA_{100}$, $CdSA_{250}$, $CdMJ_{2.5}$ and $CdMJ_{5.0}$ treatments were 53, 127, 76 and 58%, respectively (Fig 2A). The $H_2O_2$ contents in leaves were boosted in all Cd-treated leaves with or without SA and MeJA supplementation in relation to control (Fig 2B). The $H_2O_2$ contents varied significantly among the treatments showing a maximum production in Cd-alone stress treatment which is 76% greater than the control. Exogenous application of SA and MeJA variably reduced this enhancement of $H_2O_2$ in green amaranth leaves depending on their respective concentrations. Compared to Cd alone treatment, the $H_2O_2$ levels were reduced by 14, 4, 12 and 21% in $CdSA_{100}$, $CdSA_{250}$, $CdMJ_{2.5}$ and $CdMJ_{5.0}$ treatments, respectively (Fig 2B). Among phytohormone treatments, supplementation of 100 μM SA and 5 μM MeJA efficiently abated Cd-mediated oxidative injury by restricting the generation of MDA and $H_2O_2$ in green amaranth plant.

## Proline content and enzymatic antioxidant activity

A remarkable alteration in leaf proline content and antioxidative enzyme activities was apparent in Cd-treated plants with or without SA and MeJA application (Fig 3). The leaf proline contents were raised by 52% and 41% in Cd alone and $CdSA_{250}$ treatments, respectively compared to the control (Fig 3A). The leaf proline content was maximum in $CdSA_{100}$ followed by $CdMJ_{5.0}$ and $CdMJ_{2.5}$ which were 33, 29 and 19% higher than the Cd alone treatment, respectively.

An extensive elevation of the SOD, CAT and GPX enzyme activities was noticed in all Cd-treated plants as compared to control plants (Fig 3B-D). In Cd-alone treatment, the SOD, CAT and GPX activities were significantly boosted by 59, 258 and 187%, respectively in relation to control. Moreover, the exogenous application of SA and MeJA showed an extent enhancement of these enzyme activities particularly in the $CdSA_{100}$ and $CdMJ_{5.0}$ treatments. For instance, the SOD, CAT and GPX enzyme activities were enhanced by 17, 51 and 34% in $CdSA_{100}$ and by 29, 70 and 55% in $CdMJ_{5.0}$ when compared with that of individual Cd treatment (Fig 3B-D). Considering all treatments, the plants treated with 5 μM MeJA displayed greater activities of these enzymes followed by the plants treated with 100 μM SA (Fig 3B-D).

## Cd contents and translocation in plant organs

The Cd contents in root, shoot and whole plant as well as Cd translocation from medium-root-shoot is presented in Fig 4. A higher level of Cd in root and shoot was found in Cd-alone treated green amaranth plants while reduced root and shoot

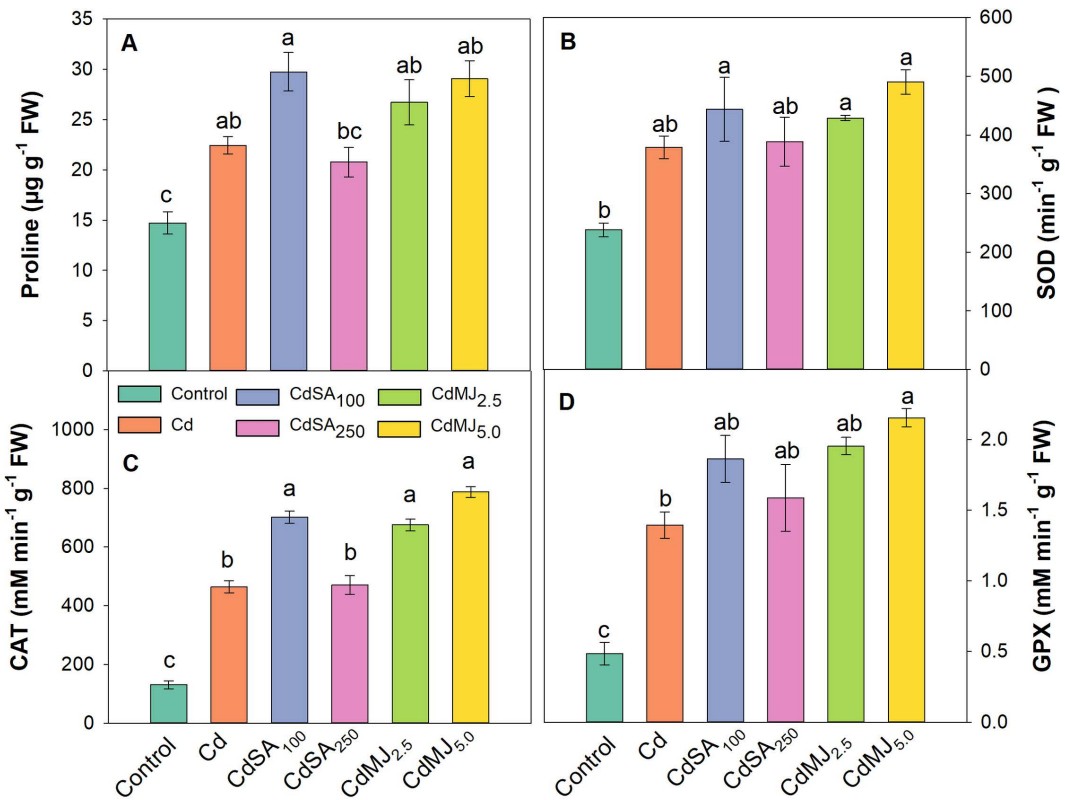

**Fig 3. Leaf proline contents (A) and enzymatic antioxidants activities: Superoxide Dismutase (SOD; B), Catalase (CAT; C), and Guaiacol Peroxidase (GPX; D) of 42-day-old green amaranth plants grown in control and Cd with or without SA and MeJA supplementation in hydroponic conditions.** The upright bars relate to SEM (n = 3). Treatment means with unlike lowercase letters within the treatments denote significant variations corresponding to Tukey's HSD test at $p < 0.05$. Treatments are the same as Fig 2.

Cd contents were recorded in phytohormone treated plants (Fig. 4A). The plants accumulated higher Cd in roots than in shoots in all treatments. Apart from control, the lowest Cd contents in root were recorded in $CdSA_{100}$ (5.72 µg g$^{-1}$ DW) and $CdMJ_{5.0}$ (5.94 µg g$^{-1}$ DW) treatments which were 30% and 27% lower than the individual Cd treatment (8.13 µg g$^{-1}$ DW), respectively (Fig 4A). The shoot Cd content was also found significantly higher in Cd-alone treatment (6.19 µg g$^{-1}$ DW) while the shoot Cd contents were reduced by 51 and 54% in the $CdSA_{100}$ and $CdMJ_{5.0}$ treatments, respectively compared to that of Cd-alone treatment. Total Cd contents greatly altered among the growth conditions and ranked as $Cd > CdSA_{250} > CdMJ_{2.5} > CdMJ_{5.0} > CdSA_{100} > Control$. Cd stress exhibited significantly higher BCF in root and shoot and the lower root and shoot BCF were observed in $CdSA_{100}$ and $CdMJ_{5.0}$ treatments, respectively (Fig 4B). A higher root BCF was apparent in comparison to shoot BCF indicating an enhanced uptake of Cd from the growth medium to root than to shoot. In Cd-alone treated plants, Cd translocation from root to shoot was higher as reflected by increased TF while the application of SA and MeJA restricted Cd translocation from root to shoot demonstrating lower TF in green amaranth plant (Fig 4B).

## Hierarchical clustering of treatments and traits

The magnitude of Cd tolerance is demonstrated by a two-way hierarchical clustering heatmap considering the STI scores of all 30 measured traits (Fig 5). The heatmap displays the clustering of treatments (row-wise) and traits (column-wise) into three categories for each. Among treatments, the Cd and $CdSA_{250}$ belong to Cluster 1 (C-I) having lower STI values (dark red in color) indicating greater sensitivity to Cd. In contrast, $CdMJ_{5.0}$ solely belongs to Cluster-II (C-II) exhibiting

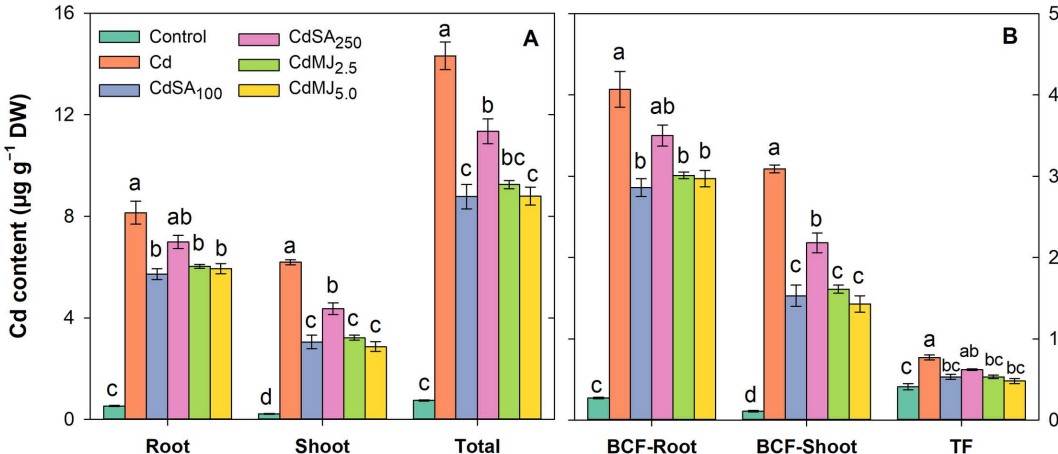

**Fig 4. Cd contents in plant organs (µg g⁻¹ DW), bioconcentration factor (BCF) and translocation factor (TF) in 42-day-old green amaranth plant grown in control and Cd with or without SA and MeJA supplementation in hydroponic conditions.** The upright bars relate to SEM (n = 3). Treatment means with unlike lowercase letters within the treatments denote significant variations corresponding to Tukey's HSD test at $p < 0.05$. Treatments are the same as in Fig 2.

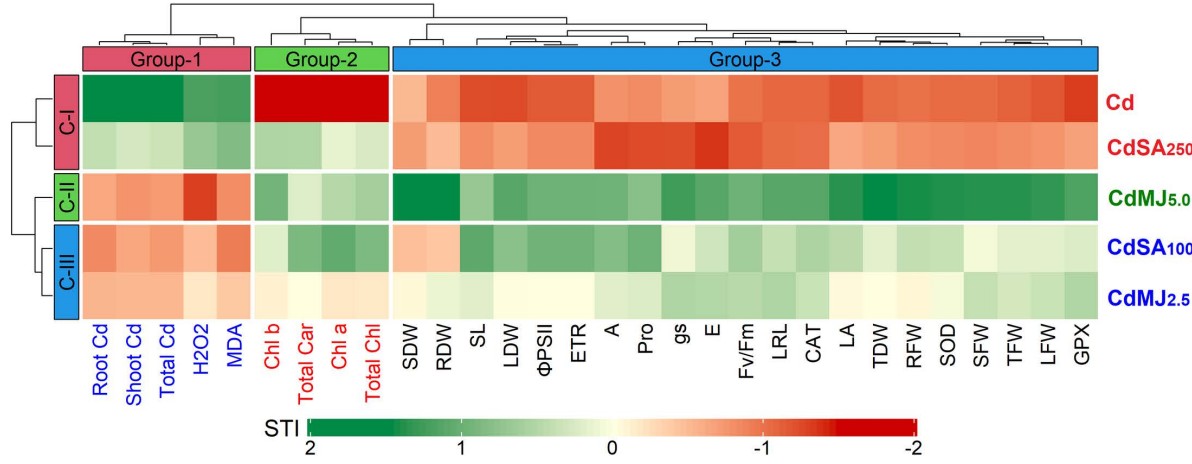

**Fig 5. Hierarchical clustering of treatments (row-wise) and measured traits (column-wise) of green amaranth plants displayed as a heatmap.** The normalized Stress Tolerance Index (STI) values were used to construct the heatmap. The STI scores are illustrated on a normalized scale of −2 (darker red) to 2 (darker green). Traits explanation: H₂O₂- Hydrogen Peroxide; MDA- Malondialdehyde; Chl b- Chlorophyll *b*; Total Car- Total Carotenoids; Chl a- Chlorophyll *a*; Total Chl- Total Chlorophyll; SDW- Stem Dry Weight; RDW- Root Dry Weight; SL- Shoot Length; LDW- Leaf Dry Weight; ΦPSII- Quantum yield of PSII; ETR- Electron Transport Rate; A- Photosynthesis rate; Pro- Proline; gs- stomatal conductance; Fv/Fm- Maximum photochemical efficiency of PSII; E- Transpiration rate; LRL- Longest Root length; CAT- Catalase; LA- Leaf Area; TDW- Total Dry Weight; RFW- Root Fresh Weight; SOD- Superoxide Dismutase; SFW- Stem Fresh Weight; TFW- Total Fresh Weight; LFW- Leaf Fresh Weight; GPX- Guaiacol Peroxidase.

greater Cd tolerance scoring maximum STI (dark green). The rest two treatments, CdSA₁₀₀ and CdMJ₂.₅ in Cluster-III (C-III) represented moderate tolerance to Cd based on STI scores (Fig 5). The traits in similar, clustered into three groups where Group-1, Group-2 and Group-3 consist of 5, 4 and 21 traits, respectively (Fig 5). In general, the values of traits in Group-2 and Group-3 were lowered due to Cd-toxicity compared to control and therefore, higher STI (dark green) of these traits were considered as Cd tolerant. On the other hand, the reverse phenomenon (lower STI but higher tolerance) was associated with Group-1 traits as their values were increased by Cd stress in comparison to control. Therefore, in

hierarchical clustering based on STI scores of the measured traits, the Cd tolerance among the treatments can be ordered as $CdMJ_{5.0} > CdSA_{100} > CdMJ_{2.5} > CdSA_{250} > Cd$ (Fig 5).

## Principal Component Analysis (PCA)

The PCA-biplot explained the closely associated traits and their contribution to total variation among the treatments studied (Fig 6). The two core Principal Components (PCs) – PC1 and PC2 were used to create the PCA-biplot due to their maximum contribution (84.6%) in total variation. The PC1, PC2 and PC3 scored eigenvalues of > 1 and cumulatively contributed 91.9% of total variations (Fig 7a-b). PCA-biplot separated the treatments based on the traits studied where the Control and Cd treatments kept a long distance while the SA and MeJA treatments remained in between control and Cd treatments (Fig 6). According to PCA-biplot, the growth and photosynthetic parameters were in opposite directions with root, shoot and total Cd contents, MDA and $H_2O_2$ contents. The antioxidants and proline contents lie between them, indicating that plant growth and photosynthesis under Cd stress increased through the application of SA and MeJA via the regulation of antioxidants and proline contents. PC1 entirely contributed 69.1% of the total variability and aligned with most of the studied traits such as root, shoot and total Cd contents, plant biomass, MDA and $H_2O_2$ contents, gas exchange and chlorophyll fluorescence traits (Fig 7c). In contrast, PC2 contributed 15.5% of total variability and the traits aligned with PC2 include CAT, GPX, SOD, Proline content and LFW (Fig 7d).

## Discussion

Plants are experiencing cadmium stress because of the copious presence of Cd in agricultural lands through several anthropogenic activities [10,61]. Increased uptake of Cd and its accumulation in plants impedes crop production and

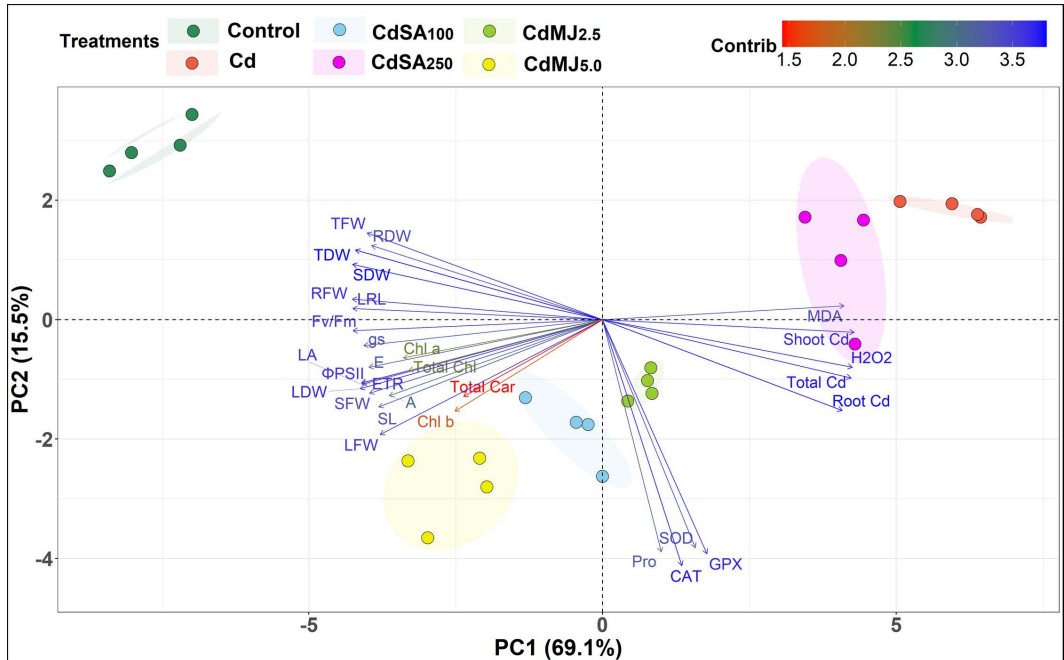

**Fig 6. PCA-biplot exhibiting the variability among six growth conditions considering 30 measured traits in green amaranth plant.** The primary two principal components (PCs) described about 84.6% (69.1% by PC1 + 15.5% by PC2) of total flexibility. The arrow length of each trait denotes the magnitude of contribution along with a color ramp from red to blue (prolonged with a darker blue arrow refers to a higher contributing trait). Traits are the same as Fig 5.

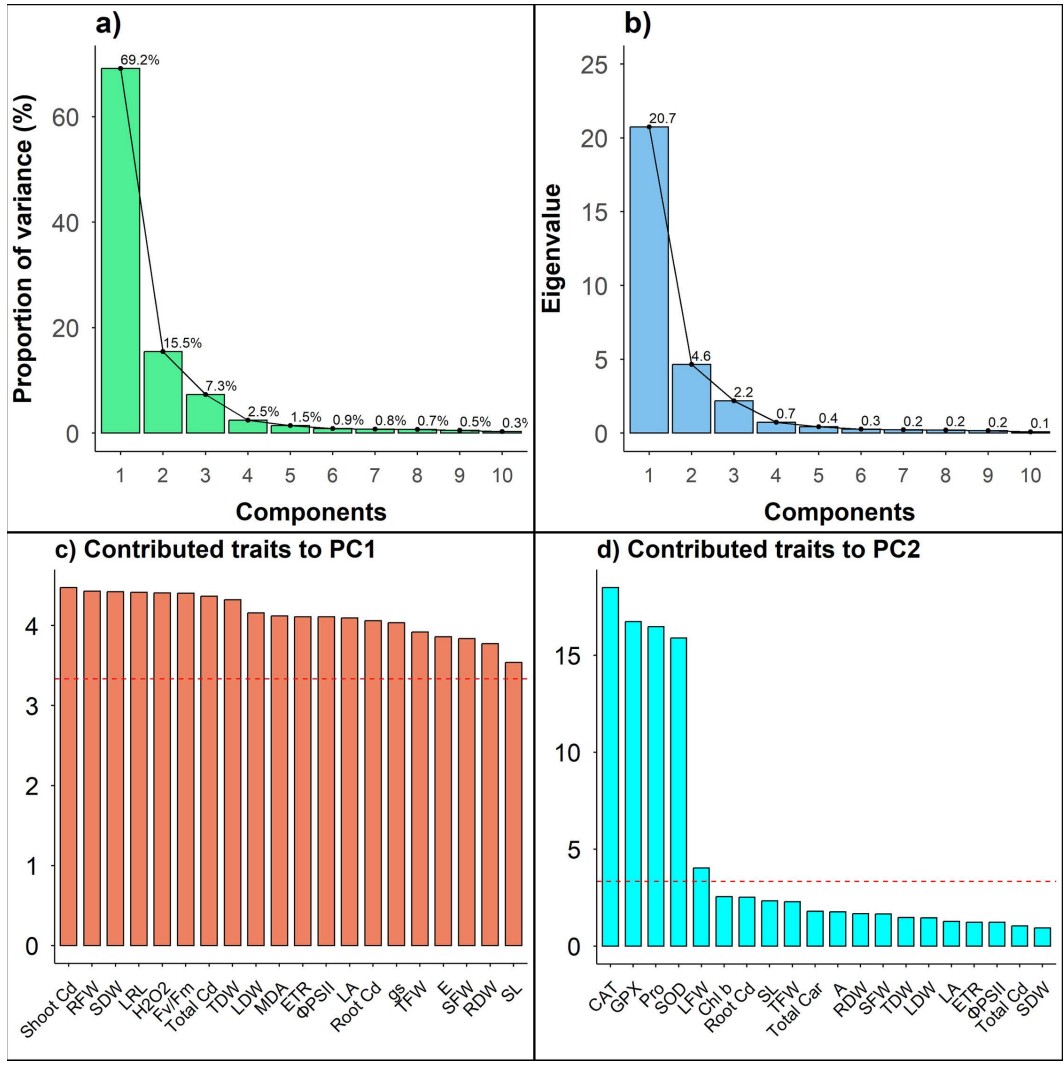

**Fig 7. Percent contribution (a) and Eigenvalues (b) of top 10 principal components (PCs) extracted from the PCA-biplot.** The first 20 contributed traits to PC1 **(c)** and PC2 **(d)**. Red-dashed lines represent the threshold limit of contribution to the respective PC.

facilitates public health risk [5,14]. Phytohormones, like salicylic acid and methyl jasmonate, are strong signalling molecules and play an imperative role in protecting plants from heavy metal stress by altering morpho-physiological and biochemical attributes [22,29]. This study assessed the efficacy of exogenous SA and MeJA application to alleviate Cd toxicity in green amaranth plants and to elucidate the underlying mechanisms of these phytohormones in improving Cd-mediated inhibition of growth, physiology and redox regulation.

In the current study, Cd stress substantially restricted the growth of green amaranth plants in terms of shoot and longest root length, leaf area and plant biomass while both SA and MeJA supplementation noticeably attenuated this growth inhibition. Cd-mediated restriction of meristematic cell division, obstruction of the parenchyma enlargement and the damage of cortical, endodermal and pericycle regions of root cells would cause amaranth growth inhibition in this study [62,63]. Further, limited progression of cell cycle and irreversible alteration of the proton pump due to Cd stress may disrupt cell membrane permeability which would hinder essential ions (N, P, K) absorption and thereby, restrict plant

growth and biomass in green amaranth [64]. SA and MeJA may help in controlling other growth-promoting hormones such as auxin, gibberellin and ethylene promoting cell division, elongation and differentiation which are essential for biomass accumulation [65,66]. MeJA application increased the formation of adventitious roots in Cd-treated cucumber plant compared to Cd-alone treated plants due to the accelerated activation of cell cycle (promoted transition from G1 to S transition phase) through the upregulation of cell-cycle-related cyclin-dependent kinase (*CDKs*) genes [67]. The application of SA is reported to increase maize root growth promoting lateral root formation along with the increase in primary root [68]. Thus, MeJA and SA supplementation in green amaranth plant in this study might counteract Cd-induced growth inhibition by facilitating cell division, enlargement and differentiation and increasing lateral root formation. Exogenous SA and MeJA application have been reported to increase their endogenous levels that may participate in alleviating heavy metal stress [69–70]. The JA-deficient *spr2* (*suppressor of prosystemin-mediated responses 2*) mutant enhanced tomato seedlings sensitivity to Cd stress highlighting the potential role of endogenous JA in amelioration of Cd stress [71]. The increased accumulation of JA, ABA and proline due to the exogenous application of MeJA enhanced Cd tolerance in okra plants by regulating endogenous hormone metabolism, osmotic adjustment substances, photosynthesis pigment and ROS metabolism [46]. Chao et al. [72] reported that SA pretreatment in rice under Cd stress enhanced the expression of *OsWRKY45* (a SA responsive gene) resulting in an increased endogenous SA content and prevented membrane damage. Zheng et al. [73] extensively studied the role of endogenous SA in response to Cd stress in *Monochoria korsakowii* and claimed that endogenous SA plays crucial role in preventing photosynthetic apparatus, upregulating enzymatic antioxidants activities, enhancing the efficiency of the ASA-GSH system and protecting osmoprotectants, thereby improving plant tolerance to Cd stress. The findings in this study regarding Cd-induced growth inhibition and SA or MeJA-mediated growth restoration corresponded favorably with those observed in earlier studies including leafy vegetables [31,58,74–76].

Cd exposure extensively interrupted photosynthetic apparatus declining photosynthetic pigments and leaf gas exchange attributes in green amaranth plants in this study. Chlorophyll synthesis may be hindered directly by Cd stress due to the active retention of $Cd^{2+}$ ions in the thylakoid membrane and stroma of the chloroplast [77]. In addition, Cd stress may induce perturbation of $Mg^{2+}$ and $Fe^{2+}$ uptake disturbing chlorophyll biosynthesis cycle and ROS-mediated pigment degradation [78,79]. The damage of the PSII was reflected by the lower $F_v/F_m$, $\Phi_{PSII}$ and *ETR* showing lesser activity of PSII and lack of electron transfer from PSII to PSI. Cd hinders PSII photoactivation through binding with essential $Ca^{2+}$ sites and reduces the activity of light-harvesting complexes which are responsible for light absorption and transfer to the PSII reaction center and thereby impairs the rate of electron transport limiting the synthesis of ATP and NADPH [80]. The degraded photosynthetic pigments along with the impairment of the light-harvesting complexes and both photosystems, therefore limited the rate of photosynthesis (*A*) in Cd-treated green amaranth leaves in this study. The *A* reduction was also attributed to the lower $g_s$ indicating restricted access to $CO_2$ into the mesophyll tissues of the green amaranth plants. The application of SA and MeJA subsequently maintained chlorophyll fluorescence and leaf gas exchange parameters nullifying the decline of *A* and stomatal limitation due to Cd toxicity. MeJA has been shown to promote photosynthesis in plants under Cd stress by safeguarding the photosynthetic apparatus, augmenting Rubisco activity and enhancing chlorophyll biosynthesis [29,39,47]. In mustard leaves, MeJA increased S-assimilation and upregulated the generation of reduced glutathione (GSH) that protects the degradation of chloroplast ultrastructure thus helping in retaining the rate of photosynthesis [45]. SA, in contrast, plays significant role in regulating photosynthesis through increased Rubisco activity, light acclimatization, redox homeostasis as well as dipping stomatal limitation [81]. Under Cd stress, foliar application of SA retained photosynthesis by stimulating chlorophyll synthesis through upregulation of chlorophyll ester reductase activity in lettuce leaves [82]. The application of SA and MeJA might have upregulated chlorophyll synthesis by increasing the uptake of $Ca^{2+}$, $Mg^{2+}$ and $Fe^{2+}$ ions [44,83].

Cd stress in this current investigation exhibited enhanced oxidative stress in green amaranth leaves by orchestrating the overproduction of ROS ($H_2O_2$) and lipid peroxidation (MDA content) while the higher state of oxidative damage was repressed by the application of MeJA and SA under Cd stress. Upon entering heavy metals into the plant cells,

they trigger to generate excess ROS ($O^{2-}$, $H_2O_2$ and ˙OH) and these highly toxic and reactive compounds thus consequently damage cell membranes and oxidize lipids, proteins, nucleic acids and photosynthetic pigments [5,19,23,84]. The impaired PSII, blocked electron transport chain and disrupted chloroplast due to Cd toxicity may lead to the leakage of electrons from the thylakoid membrane generating excess ROS in green amaranth leaves [41,85]. Cd inhibits cell-repairing processes by disrupting DNA, RNA and enzymatic proteins and therefore, reduces cell proliferation and differentiation [86]. The enhanced formation of $H_2O_2$ subsequently stimulates the fatty acid peroxidation in the membrane leading to increased MDA content in plant cells. Tobacco cell death is reported under Cd stress due to the buildup of NADPH-oxidase in peroxisomes and intensified peroxidation of fatty acids by $H_2O_2$ resulting in higher MDA production [87].

Plants respond intensely to heavy metal-induced oxidative damage by upholding antioxidative machinery (non-enzymatic and enzymatic) and enhancing osmolyte biosynthesis [88]. Proline is an effective osmolyte that functions as an antioxidant and metal chelator imparting tolerance to plants against heavy metal tolerance [89,90]. Proline accumulation was higher in Cd-treated green amaranth plants particularly in SA and MeJA treatments in this study which is aligned with other previous studies [31,38,44]. The higher proline accumulation due to SA and MeJA supplementation could balance the osmotic potential of plants [91], stabilize protein synthesis and protein structure prevent metal-induced denaturation of protein [92] and maintain antioxidative enzyme activities neutralizing free radicals [93]. The exogenous SA and MeJA application exposed enhanced activities of SOD, CAT and GPX enzymes in green amaranth leaves over Cd-alone treatments in the current investigation which is corresponded to other previous studies in different crop plants [31,44,45,76,94]. These results demonstrated clear evidence of active ROS detoxification alleviating oxidative injury by SA and MeJA application in green amaranth leaves under Cd stress. Cd in general, does not contribute directly to cellular redox reactions, it indirectly elevates the production of ROS by limiting the electron transport, disrupting the antioxidant enzyme structures and intervening with the synthesis of antioxidants [95]. SA and MeJA could therefore modulate ROS metabolism through accelerated electron transport and stabilized antioxidants structures and synthesis. Along with SOD, CAT and GPX, other crucial antioxidant such as Glutathione Reductase (GR), carotenoids and Ascorbate (AsA) contents were reported to increase under Cd stress due to the application of SA and MeJA that might have improved the tolerance of green amaranth leaves to Cd toxicity by modulating the photosynthetic electron transport, membrane stability, redox homeostasis and antioxidant defense pathway such as AsA-GSH [36,39,68,96,97]. SA and MeJA have been shown to upregulate the expressions of oxidative defense-related genes improving the withstand of plants to Cd stress [98,99]. MeJA application boosted Cd-tolerance by restricting oxidative injury through the modulation of ASA-GSH cycle (involved in scavenging of $H_2O_2$), calcium and *MAPK* (mitogen-activated protein kinase) signalling in okra and pigeon pea plants [46,76]. SA-mediated potato plants under Cd stress exhibited an enhanced expression of selected genes such as *StSABP2, StSOD* and *StAPX* [70], whereas rice plants pointed to an increased expression of *OsWRKY45* gene preventing membrane damage by reducing $H_2O_2$ levels [72]. Hediji et al. [100] reported that endogenous NO and $H_2S$ increased by SA application stimulating glutathione regeneration mechanisms like glutathione reductase, S-transferase, S-nitroso glutathione reductase and glyoxalase aggravating Cd tolerance in bean seedlings. Mustard plants exposed to Cd with MeJA enhanced GSH and Sulphur (S)-assimilation escaping damage to the membrane structures and functions and thus improved Cd tolerance [45]. MeJA application scavenged ROS regulating the upraised production of antioxidative enzymes in wheat [48], mustard [101], menthol mint [39], tomato [40] and bean [38].

The uptake of Cd by root and its translocation from root to shoot were substantially altered among the treatments in this study. SA and MeJA-treated green amaranth plants showed lower Cd contents in plant organs and lesser translocation (indicated by lower TF) of Cd from root to shoot compared to individual Cd-treated plants. Several interactions regarding minerals uptake and accumulation exist among plants and Cd competes with important essential mineral nutrients following the same membrane transporters and interferes with the ionic homeostasis [102]. As a result, Cd inhibits the uptake of vital nutrients and enters the root cells through numerous cations such as $Ca^{2+}$, $Zn^{2+}$, $Fe^{2+}$, $Mn^{2+}$, and $Mg^{2+}$ transporters [103]. SA and MeJA treatments in this study might reduce the entry of Cd into the roots by facilitating the absorption of

these cations reflecting lower BCF in both root and shoot in green amaranth plants compared to individual Cd-treated plants. The plant cell wall is the first barrier against Cd uptake and limits the translocation of Cd into the cytosol through Cd sequestration by the cell wall polysaccharides [104]. Wei et al. [40] reported that MeJA application accelerates the binding, precipitation and crystallization capabilities of cell wall and deposits higher Cd in the cell wall than in the cytoplasm in the root. The authors suggested that MeJA enhances the synthesis of cell wall polysaccharides and negatively charged Cd-binding groups maintaining increased Cd in cell wall and limited Cd translocation. The cytoplasm, in addition, contains sulfur-rich ligands that interact with Cd and store the Cd-complexes in the root vacuole and therefore, functioning as a second line defense in plants [105]. MeJa has been shown to downregulate the expression of *MT1* and *KoMT2* genes which are responsible for Cd uptake and translocation and improved Cd tolerance in *Kandelia obovata* and pigeon pea plants [69,76]. SA application in similar, increased lignin biosynthesis to strengthen the cell wall and restrict Cd from entering the root cells in rice [106]. In addition, NO might be involved in SA-induced pectin synthesis, pectin demethylesterification and lignin biosynthesis as a downstream signaling molecule, contributing to reduced Cd accumulation in Cd-stressed rice seedlings. In Cd-treated rice plant, the expression of a metal transporter gene, *OsNRAMP2* was downregulated whereas the expressions of *OsPCS1* (involved in HM tolerance) and *OsHMA3* (involved in restricting Cd translocation) genes were upregulated by the application of SA [107]. Thus, the lower BCF (Cd uptake) and TF (Cd translocation) through the supplementation of SA and MeJA in green amaranth plants in this study might be associated with the stimulated uptake and translocation barriers, enhanced Cd sequestration by the root cell and cytoplasmic vacuoles, as well as the upregulation of the heavy metal tolerant gene expressions [108,109]. The SA and MeJA-mediated reduced translocation of Cd to other core cell organelles such as chloroplast, mitochondria and endoplasmic reticulum promoting the plant's normal physiological functions and thereby, improving Cd tolerance in green amaranth plant. Previous reports also indicated a restricted translocation of Cd from root to shoot in different leafy vegetables suggesting enhanced Cd sequestration in the cell wall and vacuoles and accelerated chelation of Cd with other binding ions [22,110,111].

Hierarchical clustering of the treatments and measured traits was performed to obtain a clear understanding of the responses of various traits to Cd exposure and the extent of Cd tolerance in phytohormone-treated plants. Considering all treatments, 100 µM SA and/or 5 µM MeJA application outperformed denoting that these concentrations effectively alleviated the toxic effects of Cd in green amaranth plants. Similar findings were observed in a recent study of spinach plants [22]. However, SA supplementation with 250 µM in this study did not improve Cd tolerance in green amaranth plants indicating an overdose of SA that inhibited plant growth. A high concentration of SA can be toxic to plants as it can cause tissue injury by inducing oxidative stress via excess ROS formation [112,113]. Thus, it supposes that the optimum level of phytohormone is crucial for the successful amelioration of Cd stress and the appropriate level is species-specific. As this study measured diverse traits, a PCA-biplot was executed to determine the key characters for selection in future breeding programs. The most important characteristics identified by the PCA-biplot were organ-based Cd contents, plant biomass, $H_2O_2$ and MDA contents, PS-II efficiency and antioxidative enzyme activities.

This study optimized the levels of SA (100 µM) and MeJA (5 µM) supplementation for the effective remediation of Cd-induced adverse effects in green amaranth plants under a hydroponic system. However, to apply this strategy for large-scale farming of green amaranth in Cd-contaminated field soils, further field trial is mandatory for validating the usefulness of the studied phytohormones with the optimized levels. Moreover, the appropriate methods of exogenous application such as seed priming and foliar application (spray volume and intervals, developmental stages) need to be properly examined including the cost-benefit ratio. Pot experiments with artificial Cd-polluted soils with known concentrations are also vital to endorse the sole effect of Cd as the affected areas might be contaminated with other toxic heavy metals.

## Conclusions

Cd stress substantially impaired growth and physiological properties and promoted oxidative damage with higher Cd content and translocation in green amaranth plants. Exogenous SA and MeJA application in Cd-treated green amaranth

plants stimulated growth, photosystem-II efficiency and rate of photosynthesis; and attenuated oxidative damage by reducing MDA and $H_2O_2$ accumulation and by enhancing SOD, CAT and GPX enzyme activities. Moreover, Cd uptake and its translocation from medium via root to shoot were significantly declined by SA and MeJA supplementation. The results suggest that the alleviation of Cd-toxicity in green amaranth plant by SA or MeJA application is dose-responsive and conclude that exogenous application of 100 µM salicylic acid or 5 µM methyl jasmonate could effectively mitigate the adverse effects of Cd in green amaranth plants. The study provides fundamental and useful findings that can be largely used to alleviate the toxic effects of Cd in green amaranth plants in Cd-contaminated areas after further pot and field trails. A few characters such as Cd contents, plant biomass, oxidative damage and antioxidant activities can be used as selection criteria in crop improvement programs in relation to cadmium tolerance.

## Acknowledgments

The authors sincerely express their heartiest gratitude to Md. Ashraful Alam Akash, Shaila Akter, Atia Iffat, Talha Zubayer and Md. Tanjir Rifat for their help and coordination during the experimentation.

## Author contributions

**Conceptualization:** Md. Sabibul Haque.

**Data curation:** Md. Fazle Rabbi.

**Formal analysis:** Md. Sabibul Haque.

**Funding acquisition:** Md. Sabibul Haque.

**Investigation:** Md. Tanveer Hussain, Md. Fazle Rabbi, Hafsa Tasnim, Md. Asiful Haque, Arnab Saha.

**Methodology:** Md. Tanveer Hussain, Md. Sabibul Haque, Md. Fazle Rabbi, Hafsa Tasnim, Md. Asiful Haque, Arnab Saha.

**Project administration:** Md. Sabibul Haque.

**Supervision:** Md. Sabibul Haque, AKM Golam Sarwar, Md. Nesar Uddin, Md. Alamgir Hossain.

**Visualization:** Md. Fazle Rabbi.

**Writing – original draft:** Md. Tanveer Hussain.

**Writing – review & editing:** Md. Sabibul Haque, AKM Golam Sarwar, Md. Nesar Uddin, Md. Alamgir Hossain.

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
