## [Decision Letter · Decision Letter 0]

PONE-D-25-15737Cadmium-induced impairment in growth, photosynthetic apparatus and redox regulation in green amaranth (Amaranthus viridis) plant attenuated by salicylic acid and methyl jasmonatePLOS ONE

Dear Dr. Haque,

Thank you for submitting your manuscript to PLOS ONE. After careful consideration, we feel that it has merit but does not fully meet PLOS ONE’s publication criteria as it currently stands. Therefore, we invite you to submit a revised version of the manuscript that addresses the points raised during the review process.

We look forward to receiving your revised manuscript.

Kind regards,

Debasis Mitra

Academic Editor

PLOS ONE

**Journal Requirements:**

1. When submitting your revision, we need you to address these additional requirements. Please ensure that your manuscript meets PLOS ONE's style requirements, including those for file naming. The PLOS ONE style templates can be found at https://journals.plos.org/plosone/s/file?id=wjVg/PLOSOne_formatting_sample_main_body.pdf and https://journals.plos.org/plosone/s/file?id=ba62/PLOSOne_formatting_sample_title_authors_affiliations.pdf 2. Thank you for stating in your Funding Statement: The research was conducted from a research grant supported by the Ministry of Education, the People’s Republic of BangladeshGrant ID: LS20222142; 2022/17/MoE).  Please provide an amended statement that declares *all* the funding or sources of support (whether external or internal to your organization) received during this study, as detailed online in our guide for authors at http://journals.plos.org/plosone/s/submit-now.  Please also include the statement “There was no additional external funding received for this study.” in your updated Funding Statement. Please include your amended Funding Statement within your cover letter. We will change the online submission form on your behalf. 3. Thank you for stating the following in the Acknowledgments Section of your manuscript: The authors are thankful to the Ministry of Education, People’s Republic of Bangladesh for financial support (Project ID: LS20222142; 2022/17/MoE). The authors sincerely express their heartiest gratitude to Md. Ashraful Alam Akash, Shaila Akter, Atia Iffat, Talha Zubayer and Md. Tanjir Rifat for their help and coordination during the experimentation. We note that you have provided funding information that is not currently declared in your Funding Statement. However, funding information should not appear in the Acknowledgments section or other areas of your manuscript. We will only publish funding information present in the Funding Statement section of the online submission form. Please remove any funding-related text from the manuscript and let us know how you would like to update your Funding Statement. Currently, your Funding Statement reads as follows: The research was conducted from a research grant supported by the Ministry of Education, the People’s Republic of BangladeshGrant ID: LS20222142; 2022/17/MoE). Please include your amended statements within your cover letter; we will change the online submission form on your behalf. 4. In this instance it seems there may be acceptable restrictions in place that prevent the public sharing of your minimal data. However, in line with our goal of ensuring long-term data availability to all interested researchers, PLOS’ Data Policy states that authors cannot be the sole named individuals responsible for ensuring data access (http://journals.plos.org/plosone/s/data-availability#loc-acceptable-data-sharing-methods). Data requests to a non-author institutional point of contact, such as a data access or ethics committee, helps guarantee long term stability and availability of data. Providing interested researchers with a durable point of contact ensures data will be accessible even if an author changes email addresses, institutions, or becomes unavailable to answer requests. Before we proceed with your manuscript, please also provide non-author contact information (phone/email/hyperlink) for a data access committee, ethics committee, or other institutional body to which data requests may be sent. If no institutional body is available to respond to requests for your minimal data, please consider if there any institutional representatives who did not collaborate in the study, and are not listed as authors on the manuscript, who would be able to hold the data and respond to external requests for data access? If so, please provide their contact information (i.e., email address). Please also provide details on how you will ensure persistent or long-term data storage and availability.

Reviewers' comments:

Reviewer's Responses to Questions

**Comments to the Author**

1. Is the manuscript technically sound, and do the data support the conclusions?

Reviewer #1: Yes

Reviewer #2: Yes

2. Has the statistical analysis been performed appropriately and rigorously? 

Reviewer #1: Yes

Reviewer #2: Yes

3. Have the authors made all data underlying the findings in their manuscript fully available?

Reviewer #1: Yes

Reviewer #2: Yes

4. Is the manuscript presented in an intelligible fashion and written in standard English?

Reviewer #1: No

Reviewer #2: Yes

5. Review Comments to the Author

**Reviewer #1:**  The work is interesting and may have practical implications. I would suggest authors improve the conclusions and add strong suggestions for other studies:

-The sentence starting with "Keywords;" should end with a period instead of a semicolon.

-There are abbreviations which need elaboration

-There is need to add hypothesis in the introduction.

-Please be consistent for general name of plants or scientific names throughout the article

- Scientific names of plant species must in italics. Check it throughout the manuscript.

-Sometime methods have not been fully explained and sometime proper references are missing in this section

-Ensure consistency in terminology and notation throughout the results section.

-Properties of soil especially initial soil selected properties can be added in one or two sentences in the article. Sometime sentences are too short in length. Check this thoroughly in the whole article.

-Treatment application method is not clear

-Make sure that each figure has a clear and concise caption or legend. This should briefly explain the key findings or context of the figure for readers who may not closely examine the results section.

-There are many subheadings in this section and most of the subheadings are lengthy

-Sentence structure is very poor in this section in results section

-In some instances, it would be beneficial to explain how the cited studies are relevant to the current study in discussion section.

Discussion should be rewritten to explain mechanistic explanation rather than superficially covering the topic

-Conclusion is too short.

-It's valuable to include a paragraph on the potential applications of the findings and suggest directions for future research. For example, how could this knowledge be applied in agriculture, and what specific experiments or investigations might be warranted?

-Figure looks properly prepared except missing legends in few figures

Kindly add following references to make MS more catchy

https://link.springer.com/article/10.1007/s12210-018-0702-y

https://doi.org/10.1007/978-3-030-27165-7_7

https://doi.org/10.1007/s00344-018-9854-3

https://doi.org/10.1007/s12298-019-00715-y

https://doi.org/10.3389/fpls.2022.895427

**Reviewer #2:**  Thanks to the authors for this manuscript. This is a well-written and informative piece of work. The introduction clearly establishes the context, problem, and purpose of the study. The methods sections is well-detailed and crafted, although some sentences are a bit long and could be slightly shortened for better readability (I recognize this is subjective). The discussion is well done and coherent with the results, although some phrases might be a bit speculative and I would either expand them with more information on the underlying mechanisms that support the considerations, or tone them down.

The manuscript could be published as it is, since it is of good quality, but I have some suggestions for minor improvements if the authors accept them:

"42 Increased population growth in Bangladesh

43 forces rapid industrialization and imprudent use of agrochemicals (heavy metal-containing

44 pesticides and fertilizers) "

I agree that the "imprudent use" aspect is important to emphasize, but it's not inherent to population growth. It might be more accurate to say that the increased demand for food production drives agrochemical use, which might be or not abused depending on various aspects (for example, local laws).

"45 Heavy metals incorporate into the food chain

46 through plant uptake via soil-root-crop or water-root-crop pathways triggering serious health

47 hazards for living organisms (4,5).

I would slightly rephrase this saying that heavy metals *can* incorporate (depending on the plant species, the metal, and soil conditions) and *might* trigger health hazards, also depending on amounts.

"A well-known

112 cultivated variety “Gheekanchan” from Lal Teer Seed Ltd. was chosen for this experiment."

Can you provide a brief justification for choosing this specific variety? e.g. is it known for its Cd accumulation capability, its popularity among local farmers, or some other relevant characteristic? Even if it is just because it was simple to get and manipulate.

Also, I suggest to invert the order of words as "For this experiment, a well-known etc." (stylistic choice)

"amaranth seeds were disinfected with Vitavax-200"

Can you provide a short description of this product? In particular, describing why you chose this one in particular, with which some scholars might be unfamiliar.

"116 (95%) solution was purchased from Sigma-Aldrich, Germany and 11.8 µL was used to prepare a 1

117 mM stock solution for 50 mL volume. The SA (Sigma Aldrich, Germany) stock solution (1 mML

118 -1) was made by dissolving 138.2 mg in 2 mL methanol and then adding distilled water as needed"

Could you verify thins? Is 11.8 µL of 95% MeJA in 50 mL truly 1 mM? Or should it be a different volume? Is 138.2 mg/L of SA correct, given the molecular weight (molar mass)?

"Fourteen seedlings were supported by a perforated

126 cork sheet in each tank and regarded as a single replicate. "

I suggest to expand this for researchers trying to replicate the study or use it as a base for their projects, so that they will be able to handle with flexibility the given info. Only fourteen seedlings in total might be considered a bit little for accurate statistics, while putting them in just one sheet might be a bit crowded (depending on the size of the sheet). Could you add some text specifying why the outcome should not be compromised and is well reliable?

"Cd stress was initiated in

131 hydroponic conditions (21 d old seedlings) by introducing 10 µM CdCl2·H2O in the nutrient

132 solution in all treatments except control and the duration of Cd exposure was three weeks."

I suggest to specify why you chose this specific cadmium concentration and why exactly 21 days seedlings, even if you just found it to be a practical and workable setup. You later say "The Cd concentration was chosen based on a prior trial.", is it published data that you could cite, or just internal screening? Could you give further info? This can be useful for other researchers wanting to replicate the experiment who might discuss whether to follow by the book the protocol or slightly alter it depending on the situation.

"The composition was kept in the dark to initiate the reaction and then the absorbance was recorded at 390 nm wavelength."

You mean, immediately put in the dark and then you measured absorbance?

"MeJA and SA 424 supplementation might counteract growth inhibition by facilitating cell division and enlargement 425 and avoiding cell damage."

Can you describe the known mechanisms of action for this and similar passages? This can be useful for some latter phrases which hypothesize possible consequences from it.

"Under Cd stress, MeJa may help in controlling other growth-promoting

426 hormones such as gibberellin and ethylene enabling plant growth and development (63)."

I suggest to replace "enabling" with "promoting".

"The supplemented MeJA and SA may have increased their

429 endogenous levels that would restore cell permeability modulating nutrient uptake, gene expression

430 and redox balance under Cd stress (65)"

This is a bit speculative, since you didn't measure that. I would rephrase in more cautious words, and give a possible mechanism supporting this idea.

"In addition, Cd stress may induce perturbation" [...] "The degraded photosynthetic pigments along with the impairment of the light-harvesting complexes

439 and both PSI and PSII under Cd stress may have resulted in a declined rate of photosynthesis (A)"

As above, these phrase could be a bit speculative. I recognize that they can be legitimate, so take my suggestion as a way to make your conclusions stronger and clearer for the reader.

6. PLOS authors have the option to publish the peer review history of their article (what does this mean? ). If published, this will include your full peer review and any attached files.

**Do you want your identity to be public for this peer review?** For information about this choice, including consent withdrawal, please see our Privacy Policy .

Reviewer #1: No

Reviewer #2: No

---

## [Author Response · Author response to Decision Letter 1]

30 Jun 2025

Dear Editor,

We hereby send you the revised manuscript (PONE-D-25-15737) entitled ‘Cadmium-induced impairment in growth, photosynthetic apparatus and redox regulation in green amaranth (Amaranthus viridis L.) plant attenuated by salicylic acid and methyl jasmonate’. We have taken the reviewers’ comments into account and revised them accordingly. The revised manuscript highlighting the changes made to the original version is uploaded as ‘Revised Manuscript with Track Changes’. In addition, an unmarked version without tracked changes is uploaded as 'Manuscript’. The point-by-point reviewers’ responses were made considerably and uploaded as ‘Response to Reviewers’.

Figures are updated using the digital diagnostic tool PACE as recommended.

In response to journal requirements, we have followed the styles and guidelines according to the journal template. The ‘Funding’ section is amended and edited as ‘The entire research was conducted from a research grant supported by the Ministry of Education, the People’s Republic of Bangladesh (Project ID: LS20222142; 2022/17/MoE). There was no additional external funding received for this study’.

The funding information is deleted from the acknowledgment part and kept as ‘The authors sincerely express their heartiest gratitude to Md. Ashraful Alam Akash, Shaila Akter, Atia Iffat, Talha Zubayer and Md. Tanjir Rifat for their help and coordination during the experimentation’.

For external minimal data access, please contact the institutional body ‘Bangladesh Agricultural University Research System (BAURES) and the contact address is director.baures@bau.edu.bd.

Response to Reviewers’ Comments for the Manuscript (PONE-D-25-15737)

Response to comments of Reviewer #1

Reviewer Comments:

− The work is interesting and may have practical implications.

− I would suggest authors improve the conclusions and add strong suggestions for other studies:

Authors’ response:

− We would like to express our sincere thankfulness for your appreciation and efforts spent reviewing this paper, which will undoubtedly improve the manuscript's exposure and quality. We have carefully considered your suggestions and corrected them accordingly.

− The conclusion is improved and suggestions for future studies are now added in both the discussion and conclusion parts (Lines 700–708, 720–727).

Reviewer Comments:

− The sentence starting with "Keywords;" should end with a period instead of a semicolon.

− There are abbreviations which need elaboration

− There is need to add hypothesis in the introduction.

− Please be consistent for general name of plants or scientific names throughout the article

− Scientific names of plant species must in italics. Check it throughout the manuscript.

− Sometime methods have not been fully explained and sometime proper references are missing in this section

Authors’ response:

− The sentence starting with Keywords is now corrected

− All the abbreviations in the entire manuscript are checked and elaborated.

− The hypothesis of the research is now included in the introduction (Lines 107–110)

− The entire manuscript is updated and consistent with the common names of the plants

− The scientific names of plants are corrected with italic forms including the reference section

− The methodology has been thoroughly revised with more explanation of measured traits and missing and relevant references are added.

Reviewer Comments:

− Ensure consistency in terminology and notation throughout the results section.

− Properties of soil especially initial soil selected properties can be added in one or two sentences in the article.

− Sometime sentences are too short in length. Check this thoroughly in the whole article.

− Treatment application method is not clear

− Make sure that each figure has a clear and concise caption or legend. This should briefly explain the key findings or context of the figure for readers who may not closely examine the results section.

− There are many subheadings in this section and most of the subheadings are lengthy

− Sentence structure is very poor in this section in results section

Authors’ response:

− The results section is thoroughly and carefully checked. Terminology and notations are now consistent throughout the results section. Please have a look at the results section.

− We have not used soil as a medium of plant growth and therefore did not mention soil properties.

− Short sentences are elaborated and rephrased in the entire manuscript

− The treatment application method is updated and clarified (Lines 142–149)

− The figure captions are revised and made concise and informative. Missing legends are added.

− Subheadings in the results sections are now amended with shortened form

− We have carefully considered the sentence structure in the result section and revised it accordingly

Reviewer Comments:

− In some instances, it would be beneficial to explain how the cited studies are relevant to the current study in discussion section.

− Discussion should be rewritten to explain mechanistic explanation rather than superficially covering the topic

− Conclusion is too short.

− It's valuable to include a paragraph on the potential applications of the findings and suggest directions for future research. For example, how could this knowledge be applied in agriculture, and what specific experiments or investigations might be warranted?

− Figure looks properly prepared except missing legends in few figures

− Kindly add following references to make MS more catchy

https://link.springer.com/article/10.1007/s12210-018-0702-y

https://doi.org/10.1007/978-3-030-27165-7_7

https://doi.org/10.1007/s00344-018-9854-3

https://doi.org/10.1007/s12298-019-00715-y

https://doi.org/10.3389/fpls.2022.895427

Authors’ response:

− The discussion part is modified with an emphasis on relevancy/connection between cited studies and the current study.

− The discussion is revised with more mechanistic explanation where necessary (Lines 532–660, 576–580, 630–647, 656–683, 700–708).

− The conclusion is revised and elaborated with more useful findings

− The potential applications of the findings and the direction of future research are now added as a new paragraph in the discussion section (Lines 700–708) and mentioned in the conclusion (Lines 720–723).

− Missing legends in the figures are incorporated.

− The suggested relevant citations are included in the introduction and discussion part (Reference numbers – 18, 20, 31, 39, 85)

Thanks again for your valuable time and please check the revised manuscript.

Response to comments of Reviewer #2

Reviewer Comments:

− Thanks to the authors for this manuscript. This is a well-written and informative piece of work. The introduction clearly establishes the context, problem, and purpose of the study. The methods sections is well-detailed and crafted, although some sentences are a bit long and could be slightly shortened for better readability (I recognize this is subjective). The discussion is well done and coherent with the results, although some phrases might be a bit speculative and I would either expand them with more information on the underlying mechanisms that support the considerations, or tone them down.

− The manuscript could be published as it is, since it is of good quality, but I have some suggestions for minor improvements if the authors accept them:

Authors’ response:

− We wish to express our sincere gratitude for taking the time to review this manuscript. We appreciate your valuable feedback which will certainly improve the quality of the manuscript. We have respectfully considered your valuable suggestions and revised the manuscript accordingly – in particular the discussion part.

Reviewer Comments:

− "Increased population growth in Bangladesh forces rapid industrialization and imprudent use of agrochemicals (heavy metal-containing pesticides and fertilizers)".

I agree that the "imprudent use" aspect is important to emphasize, but it's not inherent to population growth. It might be more accurate to say that the increased demand for food production drives agrochemical use, which might be or not abused depending on various aspects (for example, local laws).

− "Heavy metals incorporate into the food chain through plant uptake via soil-root-crop or water-root-crop pathways triggering serious health hazards for living organisms (4,5).

I would slightly rephrase this saying that heavy metals *can* incorporate (depending on the plant species, the metal, and soil conditions) and *might* trigger health hazards, also depending on amounts.

Authors’ response:

− We agree with your points. The sentence is now revised according to your viewpoints (Lines 42–46).

− We have rephrased the mentioned sentence according to your suggestion (Lines 47–50)

Reviewer Comments:

− "A well-known cultivated variety “Gheekanchan” from Lal Teer Seed Ltd. was chosen for this experiment."

Can you provide a brief justification for choosing this specific variety? e.g. is it known for its Cd accumulation capability, its popularity among local farmers, or some other relevant characteristic? Even if it is just because it was simple to get and manipulate.

Also, I suggest to invert the order of words as "For this experiment, a well-known etc." (stylistic choice)

− "amaranth seeds were disinfected with Vitavax-200"

Can you provide a short description of this product? In particular, describing why you chose this one in particular, with which some scholars might be unfamiliar.

− "(95%) solution was purchased from Sigma-Aldrich, Germany and 11.8 µL was used to prepare a 1mM stock solution for 50 mL volume. The SA (Sigma Aldrich, Germany) stock solution (1 mM L-1) was made by dissolving 138.2 mg in 2 mL methanol and then adding distilled water as needed"

Could you verify thins? Is 11.8 µL of 95% MeJA in 50 mL truly 1 mM? Or should it be a different volume? Is 138.2 mg/L of SA correct, given the molecular weight (molar mass)?

Authors’ response:

− Thanks for the valuable point. The availability of green amaranth cultivars is limited in Bangladesh and this variety was chosen among others due to its suitability for round-the-year production, easy availability, fast and vigorous growth, attractive green leaves, softness and deliciousness. We have now added the justification for choosing this variety (Lines 119–124). Also, the sentence is reformed according to your suggestion

− A short description of the product Vitavax-200 is presented and justification for selecting this product is mentioned (Lines 124–128)

− The calculation and formulation of MeJA and SA solution preparation are correct. MeJa was purchased from Sigma Aldrich as a 95% solution. Please have a look at the following figure for the MeJA calculation in detail (please find the figure in the uploaded file 'Response to Reviewers'. The molecular mass of SA is 132.12 g/mol which means, for 1 M solution, we need to add 132.12 g to one liter of deionized water. Therefore, we added 132.12 mg to one liter which is equivalent to 1 mM with a final volume of one liter.

Reviewer Comments:

− "Fourteen seedlings were supported by a perforated cork sheet in each tank and regarded as a single replicate. "

I suggest to expand this for researchers trying to replicate the study or use it as a base for their projects, so that they will be able to handle with flexibility the given info. Only fourteen seedlings in total might be considered a bit little for accurate statistics, while putting them in just one sheet might be a bit crowded (depending on the size of the sheet). Could you add some text specifying why the outcome should not be compromised and is well reliable?

− "Cd stress was initiated in hydroponic conditions (21 d old seedlings) by introducing 10 µM CdCl2·H2O in the nutrient solution in all treatments except control and the duration of Cd exposure was three weeks."

I suggest to specify why you chose this specific cadmium concentration and why exactly 21 days seedlings, even if you just found it to be a practical and workable setup. You later say "The Cd concentration was chosen based on a prior trial.", is it published data that you could cite, or just internal screening? Could you give further info? This can be useful for other researchers wanting to replicate the experiment who might discuss whether to follow by the book the protocol or slightly alter it depending on the situation.

− "The composition was kept in the dark to initiate the reaction and then the absorbance was recorded at 390 nm wavelength."

You mean, immediately put in the dark and then you measured absorbance?

Authors’ response:

− We used a single tank as one replicate, each containing 14 seedlings. A total of three such tanks (replicates) were maintained per treatment, resulting in 42 plants per treatment group. This has now been clearly stated and elaborated in the revised methodology section (Lines 146–149). While placing 14 seedlings in a single sheet did create some crowding—especially for control plants toward the end of the experiment—we managed the setup effectively over the 42-day growth period. Nonetheless, we acknowledge this limitation and will take it into consideration in future experiments.

− Prior to selecting cadmium (Cd) treatment levels, we conducted an internal screening using green amaranth to determine appropriate concentrations. This information has now been added to the methodology section (Lines 154–158). For a practical and workable setup, seedlings were first established over a 21-day period, followed by 21 days of Cd exposure. Based on findings from our recent study and supporting literature, we hypothesized that 21 days would be sufficient to elicit measurable responses in leafy vegetables to Cd stress. Additionally, this duration was selected to align with the typical harvest period (40–45 days) for green amaranth. Further clarification on this experimental design is now included in the methodology section (Lines 158–163).

− The extract was kept in the dark for one hour before absorbance readings were taken. We have corrected the relevant sentence in the revised manuscript (Line 220)

Reviewer Comments:

− "MeJA and SA supplementation might counteract growth inhibition by facilitating cell division and enlargement and avoiding cell damage."

Can you describe the known mechanisms of action for this and similar passages? This can be useful for some latter phrases which hypothesize possible consequences from it.

− "Under Cd stress, MeJa may help in controlling other growth-promoting hormones such as gibberellin and ethylene enabling plant growth and development (63)."

I suggest to replace "enabling" with "promoting".

− "The supplemented MeJA and SA may have increased their endogenous levels that would restore cell permeability modulating nutrient uptake, gene expression and redox balance under Cd stress (65)"

This is a bit speculative, since you didn't measure that. I would rephrase in more cautious words, and give a possible mechanism supporting this idea.

− "In addition, Cd stress may induce perturbation" [...] "The degraded photosynthetic pigments along with the impairment of the light-harvesting complexes and both PSI and PSII under Cd stress may have resulted in a declined rate of photosynthesis (A)"

As above, these phrase could be a bit speculative. I recognize that they can be legitimate, so take my suggestion as a way to make your conclusions stronger and clearer for the reader.

Authors’ response:

− This part of the discussion is now revised with known mechanisms of MeJA and SA in relation to cell division, enlargement and growth stimulation under Cd stress with latest and relevant references (Lines 532–545)

− The sentence is revised according to your suggestion

− We are in full agreement with your point of view. The role of endogenous SA and MeJA in modulating plant growth and physiology is now discussed with supporting references. We have extensively modified this part of the discussion (Lines 545–560 ).

− Similar to the above comment, considering your viewpoints, we have revised this specific part accordingly (Lines 576–582). In addition, we thoroughly revised the discussion section in a mechanistic approach incorporating more recent a

---

## [Decision Letter · Decision Letter 1]

Cadmium-induced impairment in growth, photosynthetic apparatus and redox regulation in green amaranth (Amaranthus viridis L.) plant attenuated by salicylic acid and methyl jasmonate

PONE-D-25-15737R1

Dear Dr. Haque,

We’re pleased to inform you that your manuscript has been judged scientifically suitable for publication and will be formally accepted for publication once it meets all outstanding technical requirements.

Kind regards,

Dr. Debasis Mitra

Academic Editor

PLOS ONE

Additional Editor Comments (optional):

Reviewers' comments:

Reviewer #1: Thanks for the nice discussion. The authors have satisfactorily replied all my queries. Once again thank you

Reviewer #2: I thank the authors for resubmitting the manuscript and addressing the proposed points. I'm particularly appreciating the verification of the calculations and the expansion of the discussion. While it is in their right to hypothesize potential underlying causes for the results, it is methodologically safer a more conservative approach of first standing on established mechanisms of action and staying within the scopes of the study, until more specifical physiological and molecular investigations are performed. This work is in general a good result and in my judgement could be published on the journal, if the editor agrees.

The last thing that might be done could be a final check of potential small grammar mistakes, mispellings, formatting issues etc. that went unnoticed, as in many manuscripts there are often some that escape even vigilant eyes double or triple checking. But I don't want to cause distress to the authors, to whom I give my best wishes in their future research.

---

## [Editor Report · Acceptance letter]

PONE-D-25-15737R1

PLOS ONE

Dear Dr. Haque,

I'm pleased to inform you that your manuscript has been deemed suitable for publication in PLOS ONE. Congratulations! Your manuscript is now being handed over to our production team.

Kind regards,

on behalf of

Dr. Debasis Mitra

Academic Editor

PLOS ONE